# The 2018-2023 drought in Berlin: impacts and analysis of the perspective of water resources management

Ina Pohle[1], Sarah Zeilfelder[1], Johannes Birner[1], Benjamin Creutzfeldt[1]

[1]Department of Integrative Environmental Protection (II), Senate Department for Urban Mobility, Transport, Climate Action
and the Environment, Berlin, 10179, Germany

*Correspondence to*: Ina Pohle (Ina.Pohle@senmvku.berlin.de)

**Abstract.** The years 2018 to 2023 were characterised by extreme hydrometeorological conditions, with record-high average
annual air temperatures and record-low annual precipitation across large regions of Europe. Berlin, the capital of Germany is
potentially vulnerable to drought conditions due to its location in a relatively dry region with relatively high water demand and
complex water resources management in the Spree and Obere Havel catchments.To address the impacts of the 2018-2023
drought, various water resources management measures were implemented in Berlin and the Spree and Obere Havel
catchments.

As a case study of how droughts impact large cities, we analysed observed and modelled time series of hydrometeorological,
hydrogeological and hydrological variables in Berlin and the Spree and Obere Havel catchments to characterise the years 2018-
2023 also in comparison with long-term averages.

We found that the meteorological drought propagated into soil moisture drought and hydrological drought, e.g. in terms of
record-low groundwater and surface water levels and streamflow, with smaller rivers drying up. Due to the intensity and
duration of the drought, water resources management was only able to partially counteract the drought situation, so that water
use was partially limited, e.g. in terms of shipping. Enhanced proportions of sewage water and reverse flow were associated
with detectable concentrations of trace substances. However, Berlin's water supply was always guaranteed and represents a
stable system.

Climate change is expected to lead to more frequent meteorological droughts, which will have more severe hydrological
impacts in the future due to socioeconomic changes in Berlin (increasing population) and the catchments (termination of
mining discharges). Therefore, water resource management in Berlin and in the Spree and Obere Havel catchments needs to
be adapted to combat such situations, taking into account the lessons learned from the 2018-2023 drought and possible future
developments.

The integrative and multidisciplinary study can help better assess drought impacts in the Berlin-Brandenburg region and to
guide water management planning under potentially drier conditions. We suggest that the integrative approach presented here
can be transferred and adapted to study drought impacts on other large cities.

## 1 Introduction

Droughts as periods with water availability significantly below average are natural disasters affecting ecology, economy and society (Van Loon and Laaha, 2015). Drought propagates in the hydrological cycle (Van Loon, 2015) whereby drought conditions typically emerge from precipitation deficit (meteorological drought), and can subsequently lead to soil moisture deficit (agricultural drought), low flows and low water levels in surface and groundwater (hydrological drought), in consequence limiting water availability for various water users (socio-economical drought). Drought further affects water quality in terms of e.g. nutrients and trace substances (e.g. Winter et al., 2023; Wolff and van Vliet, 2021). Hence aquatic ecosystems can be under stress due to reduced water quantity and deteriorated water quality (Chow et al., 2022; Lennox et al., 2019). Managing droughts, low flows and water scarcity are therefore essential tasks of water resources management.

In Germany and throughout Europe, the 2018-2023 drought was characterised by extreme heat and precipitation deficit (Kaspar and Friedrich, 2020; Vogel et al., 2019), severe soil moisture drought (Boeing et al., 2022; Toreti et al., 2019), low groundwater levels and long and intense low flow periods (Boergens et al., 2020; Bonaldo et al., 2023; Trauth and Haupt, 2022). Various economic sectors, such as agriculture (Conradt et al., 2023), energy production (Shyrokaya et al., 2024) and water supply (Rickert et al., 2022; Schwandt et al., 2022) were strongly affected by the drought.

Berlin and the catchments of Spree and Obere Havel are especially vulnerable to water scarcity due to (i) low natural water availability, (ii) declining mining discharges from open-pit lignite mines (Pohle et al., 2019), (iii) a groundwater deficit of billions of cubic meters in the Lusatian lignite mining region and (iv) intense water usage for agriculture, industry, tourism and energy production. Low flows pose a potential risk to Berlin's water supply, which in contrast to other regions in Germany, is predominantly dependent on surface water. While 74 % of Berlin's drinking water comes from bank filtration and groundwater recharge and only 26 % directly from groundwater, nationally  63 % of drinking water is obtained from groundwater and only 16 % from bank filtration and recharge (StaBuA, 2022).

The 2018-2023 meteorological drought resulted in both severe soil moisture and groundwater deficit and affected Berlin's water quantity and water quality. Thus, targeted water management measures were carried out both in Berlin itself and the Spree and Obere Havel catchments.

Unlike floods, which are managed through legal (Floods Directive, Federal Water Act (Wasserhaushaltsgesetz WHG)) and administrative (German Working Group on water issues of the Federal States and the Federal Government, LAWA and River Basin Community (RBC) Elbe) instruments at EU, national and river basin level, comparable instruments have so far been lacking for the management of drought, water shortages and low flows. At the political level, the German Conference of the Environment Ministers (UMK) therefore decided to develop an overarching framework for managing drought risks at state level (UMK: 102. Umweltministerkonferenz am 7. Juni 2024 in Bad Dürkheim, 2025) The strategic framework was set at the German national level through the National Water Strategy (BMUV: Nationale Wasserstrategie. Kabinettsbeschluss vom 15. März 2023, 2025). The Capital Region Water Strategy 2050 for Berlin and Brandenburg and the Water Masterplan (SenUVK: Masterplan Wasser Berlin. 1. Bericht, 2025) address the issue at regional level. The topic of water resource management has

been addressed in the work programme of the German Working Group on water issues of the Federal States and the Federal Government (LAWA). For example, a LAWA subgroup on water shortage and competing uses has been formed in 2023 to develop guidelines for prioritising water use during periods of water shortage and for dealing with conflicts of use. The German Association of Towns and Municipalities has presented a handout as an example of how to limit the use of drinking water during hot summers (DStGB, 2023).

Understanding the causes and impacts of drought in Berlin is a prerequisite for assessing the consequences of a potentially warmer and drier future. This provides a basis for adapting water resource planning and management and for developing appropriate technical, legal and administrative instruments in a timely manner.

This manuscript analyses the 2018-2023 drought period in Berlin focusing on water quantity under consideration of complex water management measures in Berlin and the Spree and Obere Havel catchments. The manuscript further addresses the following questions: How did the 2018-2023 drought propagate in Berlin and how does it compare to previous drought periods? Does the 2018-2023 drought give a taste of the future? What does the 2018-2023 drought imply for future water resources management in Berlin and the Spree and Obere Havel catchments?

## 2 Study area

Berlin is the capital and largest city of Germany in terms of population (currently around 3.9 million and growing) and area (892 km²). As a city-state, Berlin is surrounded by the state of Brandenburg. Together, the Berlin-Brandenburg region has a population of about 6.2 million and covers about 30,500 km². The hydrological situation of Berlin is influenced by the meteorological processes in Berlin itself, but especially by its major tributaries, the Spree and Obere Havel. This study thus not only focuses on Berlin, but also considers the effects of the 2018-2023 drought on the Spree and Obere Havel catchments mainly in Brandenburg, also covering parts of the Czech Republic, Saxony and Mecklenburg-Western Pomerania (Fig. 1). The region, shaped by its glacial history, is characterised by predominantly sandy soils and flat topography. The dominant land uses are agriculture (ca. 43 % of the state of Brandenburg, most important crops are maize, winter wheat, rye and winter barley) and forestry (ca. 37 % of the state of Brandenburg, mainly Scots pine). Urban areas are concentrated in and around Berlin. Open-pit lignite mining significantly influences the Spree catchment with active mines, recultivation areas, and a groundwater deficit of billions of cubic meters in the Lusatian lignite mining region due to the pumping of mine water and subsequently the filling of open-pit lakes.

### 2.1 Hydrometeorological characteristics

The climate in the Berlin-Brandenburg region is humid continental with annual average temperatures around 9.7°C (reference period: 1991-2020, January mean: 0.8°C and July mean: 19.3°C) and annual precipitation around 579 mm (reference period: 1991-2020, January: 47 mm, July: 75 mm, see supplement S 1 based on (DWD: Annual regional averages of air temperature and precipitation (monthly mean): Index of /climate_environment/CDC/regional_averages_DE/annual/ [data set], 2024)). In

comparison with other German regions, Berlin-Brandenburg appears as one of the driest but also relatively warm (see S1). The region has undergone climatic changes in recent decades, such as air temperature increases (Gädeke et al., 2017). Annual precipitation did not change significantly, yet precipitation shifts from summer to winter (Murawski et al., 2016) and longer dry periods and more drought months in summer (Paton et al., 2021) have been observed. Overall, the region, which has relatively low natural water availability, has become even drier as temperature-induced evaporation increases have not been offset by changes in annual precipitation. Declining groundwater recharge as well as lower lake levels have been attributed to both climatic and land cover changes (Kaiser et al., 2015; Natkhin et al., 2012).

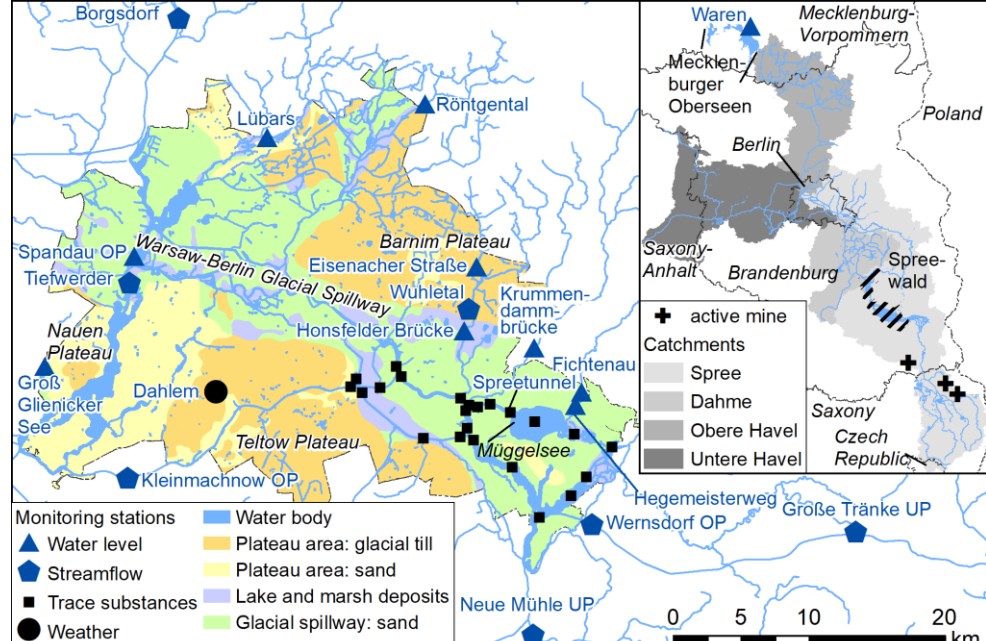

**Figure 1 Map of Berlin including the monotoring stations analysed in this study. The inset shows Berlin within the Spree (including Dahme) and Obere Havel as well as Untere Havel catchments. Data: BfG. DWD, LfU, SenMVKU, WSA**

## 2.2 Water management in the Spree and Obere Havel catchments

The Spree has its sources in Saxony; its headwater catchment also includes parts of the Czech Republic. The Spree then flows through Saxony, Brandenburg and Berlin where it joins the Havel upstream of Tiefwerder. The Spree catchment has an area of around 10,100 km²; its landscape features include large reservoirs (i.e. Bautzen, Quizdorf and Spremberg), the Lusatian

lignite mining region and the Spreewald wetland. Mining discharges from large-scale open-pit mining and reservoir management result in elevated streamflow volumes and reduced temporal streamflow variability of the Spree (Pohle et al., 2019). Open-cast lignite mining in the Spree River catchment peaked in the 1980s and declined after 1990 due to German reunification. Currently, three mines are still operating which produced 42 million tons of lignite in 2023 (DEBRIV Bundesverband Braunkohle, 2023). To extract lignite, groundwater levels have to be lowered and pumped groundwater is

released into the river system resulting in significantly elevated streamflow, i.e. up to a factor of ten during low flow periods in the 1980s (Grünewald, 2001; Koch et al., 2005). In return, decreasing mining activities and filling of post-mining lakes led to significant decreases in streamflow of the Spree since the 1990s. Another influence on the flow of the Spree are high evaporation losses in the wetland Spreewald during summer (Dietrich et al., 2007). Furthermore, water transfers between catchments (e.g. between the Spree and the Schwarze Elster and Lusatian Neiße) and abstractions (e.g. for inland fisheries)

influence streamflow variability in space and time. Water resources management in the Spree catchment is based on harmonised management principles of the working group River Basin Management Spree, Schwarze Elster and Lusatian Neisse (AG Flussgebietsbewirtschaftung Spree, Schwarze Elster und Lausitzer Neiße (AG FGB)) consisting of the German Federal States of Berlin, Brandenburg and Saxony. These management principles consider water quantity, e.g. required minimum flows (e.g. 8 m³/s for the Spree upstream of Berlin at the Große Tränke UP gauging station), and aspects of mining-

related water quality with respect to iron and sulphate concentrations. Water management is generally prioritised with water abstraction (under consideration of required minimum flow and water quality requirements) being the highest priority. Second priority is given to replenishing reservoirs, third priority to maintaining the water levels of the Oder-Spree canal and fourth priority to flooding of post-mining lakes.

The Havel River is a right tributary of the Elbe River in north-eastern Germany which originates in the state of Mecklenburg-
130 Western Pomerania upstream of the Mecklenburger Oberseen. The Havel catchment has a total area of ca. 23,900 km². Its upper part, the Obere Havel catchment, stretches up to the Spandau impoundment (gauging station Spandau OP in Fig. 1) and has an area of ca. 3,400 km². Streamflow variability of the Obere Havel is dominated by the management of the Mecklenburger Oberseen, a group of lake reservoirs (Plauer See, Petersdorfer See, Malchower See, Fleesensee, Kölpinsee and Müritz lakes) with outlets towards both the Obere Havel to the south-east and the Elde to the west. The streamflow of the Obere Havel is

135 further influenced by 14 river impoundments (Ebner von Eschenbach et al., 2021). Unlike in the Spree catchment, water resources management in the Havel catchment does not yet follow formalised management principles. The minimum flow required by Brandenburg for the Havel downstream of Berlin is 10 m³/s.

The water resources management in the Spree and Obere Havel catchments and in Berlin has been adapted to combat the 2018-2023 drought. In the Spree catchment, the working group "ad-hoc extreme situation" has been set up between the Federal

States of Berlin, Brandenburg and Saxony and the mining cooperatives to coordinate operational water management (AG FGB: Länderübergreifende Auswertung des Niedrigwassers 2018, 2019 und 2020 in den Flussgebieten Spree, Schwarze Elster und Lausitzer Neiße, 2025). These measures include limiting water abstractions, public information on water saving, optimised water management of the reservoirs and in the Spreewald wetland, adaptation of minimum flows and immission limits for

sulphate. In the Obere Havel catchment, various measures have been implemented, including a restriction of public surface-water use in some areas and an increased release from the Mecklenburger Oberseen reservoir into the Obere Havel. In Berlin, the weirs have been closed as far as possible and lock operations were limited and special monitoring programmes were set up (Creutzfeldt et al., 2021). Measures to counteract the fall in water levels in the Spandau impoundment, which is crucial for Berlin's water supply, included reduced outflows, increased transfers of waste water and suspending the targeted reduction of water levels to summer operational level (Creutzfeldt et al., 2023).

**2.3 Hydrogeology, hydrology and water management in Berlin**

Tertiary and quaternary glacial processes shaped the (hydro-) geological setup of Berlin, resulting in an aquifer system of a total thickness of approx. 150 m. This system consists of alternating layers of unconsolidated sediments with varying hydraulic conductivity such as gravel and sand, which form the aquifers and glacial till, silt and clay, which, in turn, form the aquitards. These layers can be differentiated into four aquifers (Limberg, A., Thierbach, 1997). Depending on regional sedimentation and erosion processes, some aquifers are in direct contact with each other without a separating aquitard or have been eroded by glacial-fluviatile processes. In this study, only the groundwater levels of the "main aquifer" (from the Saale glacian period) are considered, as this aquifer holds economic importance for water resource management. On the Barnim Plateau and the eastern part of the Teltow Plateau (Fig. 1), the main aquifer is confined, being covered by glacial till from the ground moraine. In the area of the Warsaw-Berlin glacial spillway, the main aquifer is covered by Weichselian sediments with high hydraulic conductivity, resulting in unconfined groundwater conditions. Also in most of the Nauen Plateau and the western part of the Teltow Plateau where the sediments of the aquifer are largely characterised by very thick meltwater sand sequences (SenSBW: Umweltatlas Berlin: Geological Outline, 2025), the groundwater is unconfined.

The general groundwater flow direction is from the recharge areas on the Barnim Plateau (in the north-east) and the Teltow Plateau (in the south) to the Warsaw-Berlin glacial spillway, which serves as the discharge area with effluent conditions to the Spree. The depth to water table ranges between 7 m and up to 40 m on the Plateau areas and only several meters along the Warsaw-Berlin glacial spillway (SenStadt: Umweltatlas Berlin: Flurabstand, 2025). Along the Warsaw-Berlin glacial spillway, groundwater levels are influenced by the Spree and its artificial regulation. The public water-supply wells are primarily located along the Spree and Lake Müggelsee in the eastern part of the city, as well as along the Havel in the western area of the Teltow Plateau. To evaluate the development of groundwater levels, it is essential to distinguish between the data sets of these (hydro-) geological regions due to the heterogeneity of the study area.

Berlin is characterised by around 140 km of larger and 75 km of smaller rivers as well as both fluvial and groundwater-fed lakes and also ponds. The main tributaries (i.e. Spree, Dahme and Obere Havel) are connected by a number of canals and regulated by weirs, resulting in a branched surface water system in Berlin. Due to weir operations, there is little intra-annual variation in water levels despite large variations in streamflow. Low streambed and water level gradients and wide river cross-sections result in near-zero flow velocities especially during drought periods. Berlin's water users, i.e. water supply and

wastewater disposal, inland navigation, abstractions and cooling water discharges by thermal power plants, depend both on the streamflow situation and the controlled water levels in Berlin.

## 3 Data and Methods

### 3.1 Data

The hydrometeorological situation was analysed using the example of the Berlin-Dahlem weather station of the German Weather Service (DWD, location see Fig. 1). We used daily values of observed precipitation and air temperature as well as derived potential evapotranspiration (grass reference evapotranspiration). We further used gridded annual air temperature and precipitation data provided by the German Weather Service for the Spree and Obere Havel catchments. Reservoirs in the Spree catchment were analysed in terms of the daily total storage of eight reservoirs; these data were provided by the Saxon Dam

Authority (LTV), Lausitzer und Mitteldeutsche Bergbau-Verwaltungsgesellschaft mbH (LMBV) and Landesamt für Umwelt Brandenburg (LfU). The Mecklenburger Oberseen in the Obere Havel catchment was considered by daily water levels at the Waren (Müritz) gauging station provided by the Federal Waterways and Shipping Administration (WSA). Inflows to Berlin were analysed based on discharges at the gauging stations Große Tränke UP (Spree), Wernsdorf OP (Oder-Spree Canal), Neue Mühle UP (Dahme) and Borgsdorf (Obere Havel) provided by WSA.

Soil moisture in Berlin has been accounted forby AMBAV (Löpmeier, 1994) model results for the Berlin-Dahlem weather station considering land cover as grass and soil as sand  between 0 cm and 60 cm depth (DWD: Calculated daily values for different characteristic elements of soil and crops. Version v19.3 [data set], 2024). Groundwater levels were analysed using daily data from monitoring wells representing four geological regions, i.e. the Nauen Plateau / Grunewald area (13 wells), the Eastern Teltow Plateau (13 wells), the Warsaw-Berlin glacial spillway (19 wells) and the Barnim Plateau (17 wells). All of the

monitoring wells are measuring the groundwater levels of the "main aquifer" which is the relevant aquifer for water resources management. The hydrological situation in Berlin was analysed using around 60 surface water quantity (water level and or flow) monitoring stations operated by WSA (https://www.pegelonline.wsv.de) or SenMVKU (https://wasserportal.berlin.de/), those which are explicitly mentioned in the manuscript are shown in Fig. 1. Data on major water users in terms of raw water abstractions, sewage water treatment plants and thermal power plants were included and obtained from the Berliner

Wasserbetriebe (BWB) and Berliner Energie und Wärme GmBH (BEW). The drought impact on Berlin's water quality was assessed by concentrations of trace substances measured at 23 sites on three sampling days in summer 2019. For an overview of the monitoring stations please refer to supplement S 1.

### 3.2 Statistical Analysis

Metrics describing the hydrometeorological situation, the hydrogeological situation and the hydrological situation as well as

water management in the drought years 2018-2023 are shown and compared to long-term averages (typically 1991-2020). If

not mentioned otherwise, hydrological years (in Germany 1 November of the previous year up to 31 October of the respective year) are used, whereby hydrological summer refers to 1 May to 31 October of the respective year.

The hydrometeorological situation is represented by thermopluviograms, relating mean annual temperature to annual precipitation totals, and evapopluviograms relating potential evapotranspiration to precipitation. The latter includes diagonals representing isolines of the climatic water balance, which is calculated as difference between precipitation and potential evapotranspiration.

To compare annual courses of the respective hydrogeological and hydrological variables of several years, we chose diagrams combining hydrographs of the hydrological years of interest (shown in colour) and percentiles for each calendar day in the reference period (shown in grey scale) based on the R code provided by Zappa et al. (2014). These figures also show long-term monthly averages. These plots may also contain hydrological main values of water level (W) or discharge (Q), i.e. highest values recorded (HHW, HHQ), highest values in a time period (HW, HQ), mean high values (MHW, MHQ), mean values (MW, MQ), mean low values (MLW, MLQ), lowest values in a time period (LW, LQ) and lowest values recorded (LLW, LLQ). Please refer to supplement S 3 for a detailed description of these hydrological main values.

The low flow situation was described using mean average minimum water level or discharge over 7 (MAM7), 30 (MAM30), 60 (MAM60) and 90 days (MAM90). Stream flow intermittency was described in terms of dry periods during which water levels drop below the threshold detectable at the water level gauges.

Groundwater levels were considered as median deviations from long-term averages for each geological region. For each well, monthly median water levels were calculated from daily values, and the monthly deviation to the long-term medians (2001-2020) of each observation well was calculated. Following from this, the monthly deviations were then averaged across all monitoring wells within each respective hydrogeological region.

Drought propagation was summarised following Peters et al. (2006) and Van Loon (2015) in terms of sequence of monthly precipitation, climatic water balance, soil moisture, groundwater levels and streamflow in relation to long-term averages.

### 3.3 Modelling

The 1 D hydronumerical model BIBER (computation and information system of Berlin's surface waters, IWU, 2019) which is based on the HYDRAX model (Oppermann et al., 2015) was used to determine streamflow at cross-sections in Berlin's surface water systems that are not monitored. BIBER considers daily values of inflows, water levels in Berlin's river impoundments, major water users (discharges of sewage treatment plants, waterworks withdrawals, withdrawals and cooling water discharges of the thermal power plants), precipitation and evaporation. The model simulates water level, discharge, flow velocity, cross-sectional area and flow volume for 1400 cross sections in Berlin's main river network. A balance adjustment is calculated for all these components. Please see supplement S 4 for a description of BIBER. Due to data availability issues, BIBER simulations were carried out for the period 2003-2022.

## 4 Results

### 4.1 Hydrometeorological conditions

The hydrological years 2018-2023 were very warm and characterised by low precipitation in Berlin and the Spree and Obere Havel catchments. At the Berlin-Dahlem weather station, the annual mean air temperature ranged between 11.0 °C and 11.2 °C (except 2021: 10.1 °C), markedly exceeding the 1991-2020 average of 9.9 °C (Fig. 2a). The warmest (2022), second (2020), fourth (2019) and fifth warmest (2018) years on record occurred in the 2018-2023 period. The third warmest year was recorded in 2017. During 2018-2023 (with the exception of 2021), the Spree and Obere Havel catchments were on average 1.1 to 1.4 °C warmer than the 1991-2020 average (standard deviation of ≤ 0.1 °C in individual years, see supplement S 5). From 2018 to 2022, annual precipitation at Dahlem ranged between 402 mm (2018, 2022) and 508 mm (2019) and was thus lower than the 1991-2020 average of 585 mm. Both 2018 and 2022 were the driest years on record at this station, while 2003 had almost as little precipitation (410 mm) . In the period of interest, only 2023 showed an above-average precipitation of 638 mm. The precipitation deficit during the 2018-2023 drought accumulated to 360 mm in the Spree catchment and 260 mm in the Obere Havel catchment (supplement S 6). Precipitation differences from the 1991-2020 period varied in space (standard deviation up to 40 mm based on 1 km² cells in each catchment, supplement S 6) and time. 2018 and 2022 were the driest years with an average precipitation deficit of 200 mm and 140 mm, respectively, in the Spree and 180 mm and 140 mm in the Obere Havel catchment. In 2021, the Spree catchment had around average precipitation and the Obere Havel catchment above-average precipitation (48 mm) while 2023 showed above average precipitation in both catchments.

The hydrological summers were characterised by above-average temperatures and (except 2023) below-average precipitation (Fig. 2b). The hydrological summers of 2018 and 2022 were characterized by highest temperatures (17.6°C and 17.0°C) and lowest and third lowest precipitation (140 mm and 183 mm, respectively, second lowest: 1999 (178 mm)) recorded at this station. Potential evapotranspiration in each year 2018-2023 exceeded the 1991-2020 average of 640 mm, with 2018 displaying the highest ETP on record (755 mm, Fig. 2c) followed by 2022, 2019 and 2003. As a result, the climatic water balance over the study period was negative, with 2018 showing the lowest CWB since records began (-352 mm, 1991-2020: -55 mm). Precipitation varied widely between months, with March 2022 being almost dry and all summer months of 2022 below average. In the past, there has been a clear alternation between years of high and low precipitation, resulting in phases of positive and negative climatic water balance (Fig. 2e). For example, 2003 was the year with the third lowest CWB on record, but was preceded by 2002 with positive CWB and 2004 with only slightly negative CWB. In contrast, the 2018-2023 drought was the longest consecutive period of negative annual CWB resulting in a strong decrease in cumulative CWB.

### 4.2 Hydrological conditions in the Spree and Obere Havel catchments

Reservoir contents in the Spree and Obere Havel catchments typically vary seasonally with higher storage volumes in summer than in winter (Fig. 3). The reservoirs in both the Spree and Obere Havel catchments were still well filled by the beginning of the 2018 hydrological summer. At the end of the hydrological years 2018, 2019 and 2020, the reservoir volumes in the Spree

catchment were lower than 25 hm³ (Fig. 3a). During the 2020 winter, the reservoirs filled to only half their capacity. The water level of the Mecklenburger Oberseen reservoir has remained below the upper operating water level of 62.35 m since March 2018. It fell below the lower operating water level of 61.80 m a.s.l (above sea level) for extended periods (September 2018 to January 2020, July 2019 to January 2020 and August 2020 to January 2021, Fig. 3b). Since 1990, the only other longer periods during which water levels fell below the lower operating water level were in 1991 (September to December) and 2003 (August to December). During two weeks in September 2019, water levels even reached the lowest operating water level of 61.60 m. From September 2018 to August 2021, the reservoir volumes of the Mecklenburger Oberseen were lower than the 90[th] percentile of exceedance compared to long-term mean values of the respective months between 1991 and 2020.

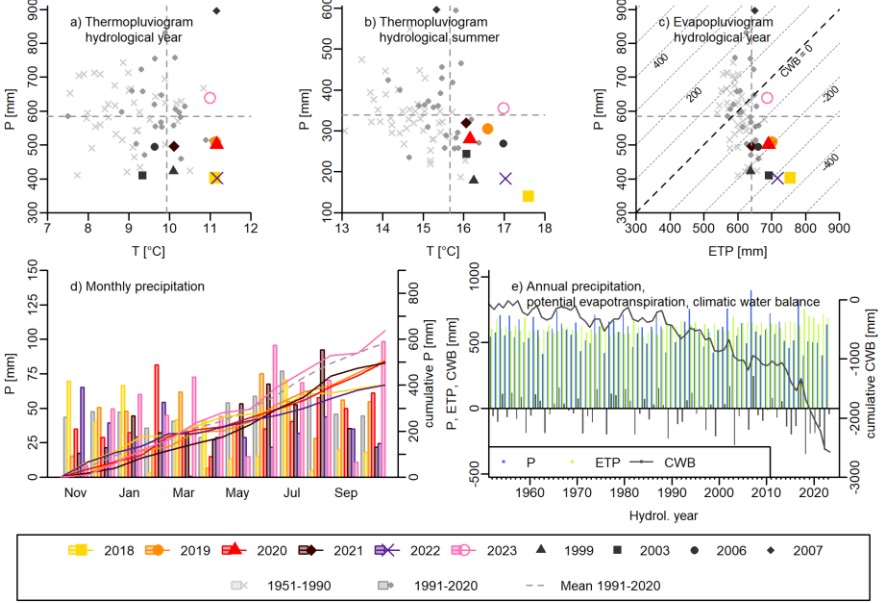

**Figure 2 Hydrometeorological situation at the example of the Dahlem weather station: a) thermopluviogram based on air temperature (T) and precipitation (P) for hydrological years, b) thermopluviogram for hydrological summer (May – October), c) evapopluviogram based on potential evapotranspiration (grass reference evapotranspiration; ETP) and P for hydrological years, d) monthly precipitation depths and cumulated precipitation for hydrological years, e) annual totals of P, ETP and climatic water balance (CWB) and cumulated climatic water balance. Data: DWD**

The inflows from the Spree and Obere Havel catchments showed extended low-flow periods (Fig. 4). The lowest annual averages on record occurred in 2019 at Borgsdorf (Obere Havel, 6.97 m³/s compared to the 1991-2020 average of 12.1 m³/s) in 2020 at Neue Mühle UP (Dahme, 2.82 m³/s compared to 8.83 m³/s) and in 2022 at Große Tränke UP (Spree, 8.56 m³/s compared to 12.0 m³/s). Compared to the reference period 1991-2020, streamflow was mostly below the respective long-term

monthly averages and below the median of the respective calendar days. During summer, streamflows were partially even lower than the 90[th] percentile of the respective calendar days, sometimes even lower than the minima in the reference period. Streamflow often fell below the average minimum flow MLQ, with the longest periods below MLQ occurring between May and September 2019 at Große Tränke UP and Wernsdorf OP, between April and September 2020 at Neue Mühle UP and between June and September 2022 at Borgsdorf. In many cases, e.g. between May and September 2018 and 2022, the flow requirement at Große Tränke UP of 8 m³/s could not be met. The most intensive low-flow periods were observed in 2019 at Große Tränke UP (MAM30, MAM60, MAM90) and in 2022 at Wernsdorf OP and Borgsdorf (all MAM values).

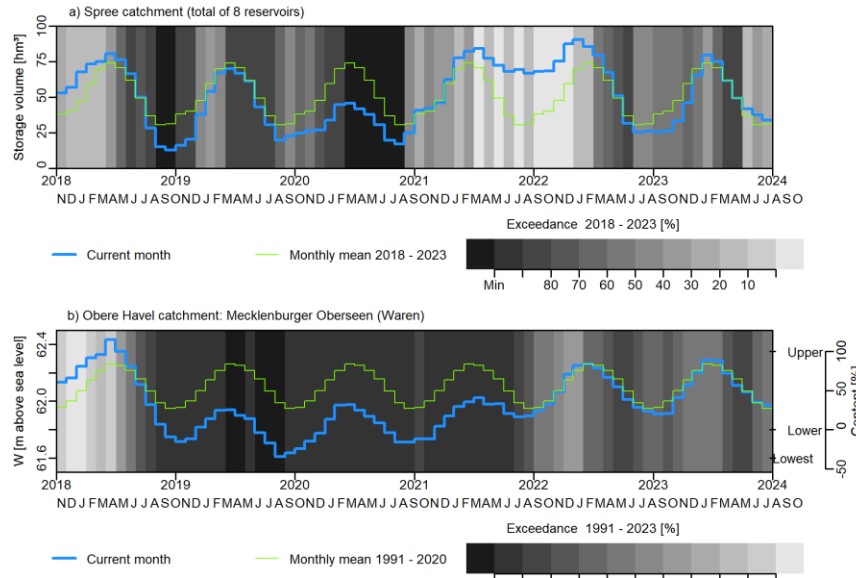

**Figure 3: Reservoir contents as monthly mean values during the 2018-2023 drought compared to long-term monthly means. a) Spree catchment: total active storage volume of eight reservoirs compared to monthly mean values in 2018-2023, b) Obere Havel catchment: water level of the Mecklenburger Oberseen at Waren gauging station compared to long-term monthly mean values in 1991-2020. The left-hand y-axis shows upper, lower and lowest operational levels and relative reservoir content. Data: LTV, LMBV, LfU, WSA**

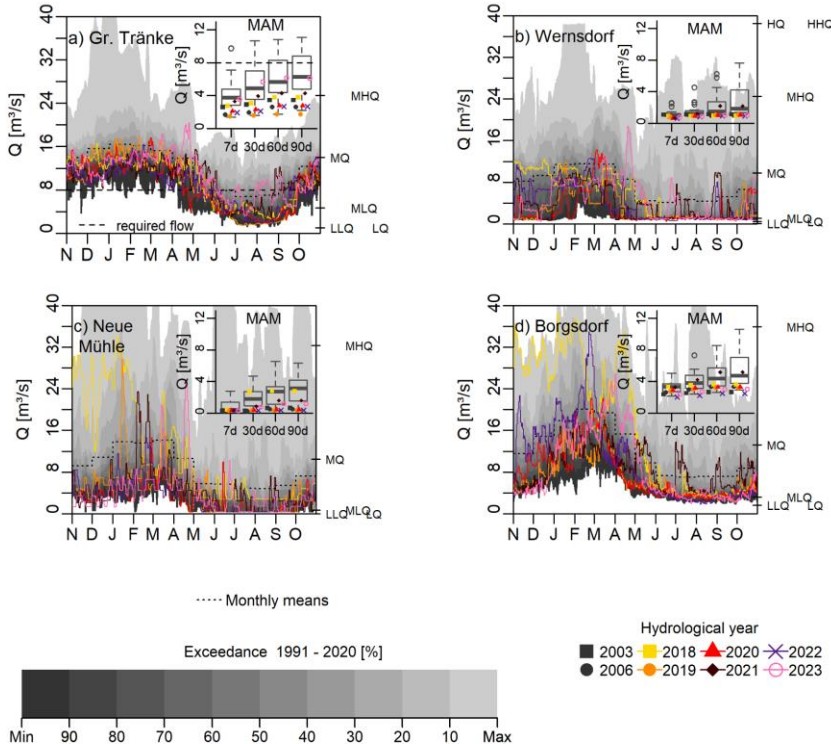

**Figure 4 Inflows (Q) to Berlin as percentiles for calendar dates and long-term monthly mean values for the hydrological years 1991-2020 and daily values in the hydrological years 2018-2023. The right-hand y-axes show hydrological main values in specific time periods: Große Tränke UP (Spree, 1962-2020, LLQ: 3 August 2001), b) Wernsdorf OP (Oder-Spree-Canal, 1962-2020, LLQ: 7th May 1905, HHQ: 6 February 2011), c) Neue Mühle UP (Dahme, 1955-2020, LLQ: February 1954) and d) Borgsdorf (Obere Havel, 1976-2020, LLQ: 29 September 2009). The insets show the low-flow characteristics MAM7, MAM30, MAM60 and MAM90 as boxplots for the hydrological years 1991-2020 and points for 2003, 2006 and 2018-2023. Data: WSA**

### 4.2 Hydrogeological and hydrological conditions in Berlin

Soil moisture shows pronounced seasonality with the highest values around 100 % of the available water capacity during January to March (Fig. 5). In the spring of 2018, soil moisture decreased to below 25 %. Since then, soil moisture above the long-term monthly averages has only been reached after rain events and in January and February. Compared to the 1991-2020 reference period, the values in 2018-2023 are generally below the median values of the respective calendar days, partially reaching or even falling below the minimum values of the reference period. The longest periods of particularly dry soils

occurred between April and December 2022 with 247 consecutive days below 50 % soil moisture for grass and sand at Dahlem.

Very low soil moisture in summer occurred also in previous years, e.g. in 2003.

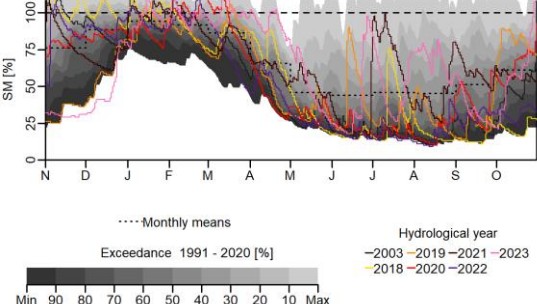

**Figure 5 Soil moisture (SM; percent of the available water capacity) for grass and sand at the Dahlem weather station shown as percentiles for calendar dates and long-term monthly mean values for the hydrological years 1991-2020 and daily values in the hydrological years 2003 and 2018-2023. Data: DWD**

At the start of the hydrological year 2018, groundwater levels were considered normal to high throughout almost all four geological regions due to heavy precipitation events in summer 2017 and the delayed reaction of the aquifers in the plateau areas (Fig. 6). Groundwater levels continued to rise until the end of the hydrological winter in 2018. However, during the hydrological summers of the years 2018 to 2020, groundwater levels decreased subsequently each year without being balanced

by a comparable rise during the groundwater recharge season of the following hydrological winter. While a seasonal pattern is recognizable in the areas of the Barnim Plateau and the Warsaw-Berlin glacial spillway, it is not evident in the eastern Teltow Plateau, indicating low groundwater recharge in that area. In the Nauen Plateau / Grunewald area, groundwater levels decreased considerably stronger than they rose during the years of 2018 to 2020. In 2021, groundwater levels showed a more balanced tendency with no major rises or declines. However, a slight decrease persisted in the eastern Teltow Plateau.

Throughout the hydrological summer of 2022, groundwater levels decreased again, especially in the Barnim Plateau due to low precipitation in the winter period of 2021/2022. Across all geological regions, groundwater levels of the analysed observation wells reached their minimum in 2022. In the subsequent hydrological winter of 2023, groundwater levels rose moderately across all regions except the eastern Teltow area. A recovery of groundwater levels occurred only in the Warsaw-Berlin glacial spillway, while in the other areas, groundwater levels remained below the range of the reference period. In 2003,

a dry year in the recent past, the groundwater levels decreased with similar extent as in 2018 although with less seasonality (except in the Warsaw-Berlin glacial spillway).

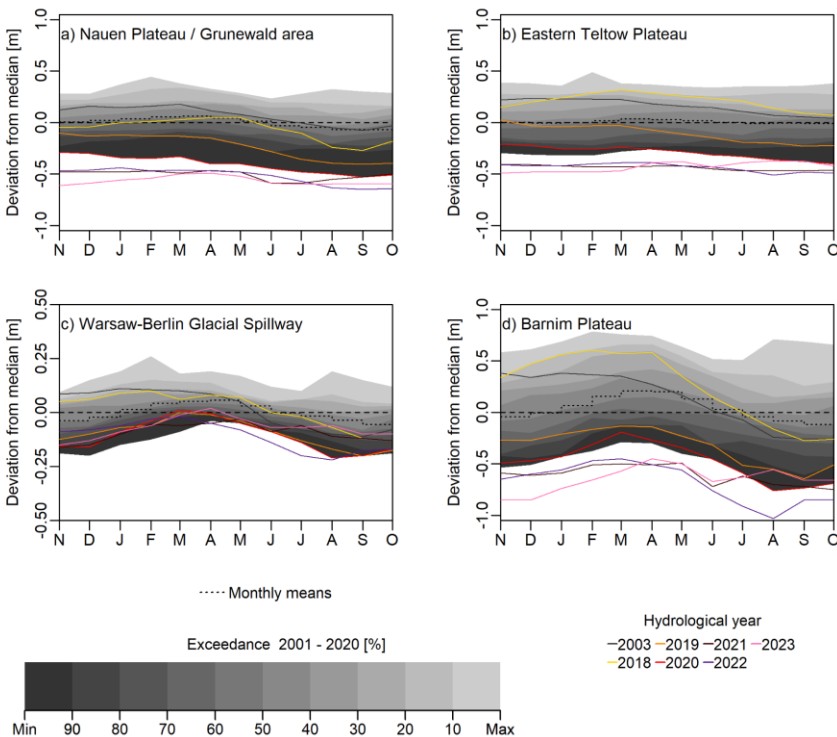

**Figure 6** Groundwater level as deviation from the long-term (2001-2020) median as percentiles for months and long-term monthly mean values for the hydrological years 2001-2020 and monthly values in the hydrological years 2003 and 2018-2023: a) Nauen Plateau / Grunewald area (median of 13 wells), b) Eastern Teltow Plateau (median of 13 wells), c) Warsaw-Berlin glacial spillway (median of 19 wells), d) Barnim Plateau (median of 17 wells). Data: SenMVKU

Groundwater-fed lakes partially recorded some of the lowest water levels since observations started, e.g. for Groß Glienicker See water levels fell between the previous LLW (lowest water ever recorded: 30.32 m in October and November 2016) so that the new LLW of 29.63 m occurred in October-November 2022 and 2023 (Fig. 7). The water level of Groß Glienicker See has been decreasing since the 1990s, with steep decline following the warm and dry years 1999 and 2003. The downward tendency has increased since 2018 with water levels generally below average low water level (MLW). Compared to the calendar months of the reference period 1991-2020, the water levels have been consistently below the 90[th] percentile of exceedance since the beginning of the hydrological year 2018.

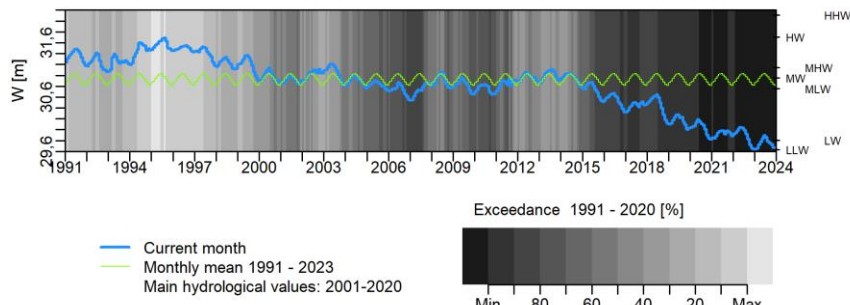

Figure 7 Lake water level (W) of the Groß Glienicker See as percentiles for months and long-term monthly mean values for the hydrological years 1991-2020 and monthly values in the hydrological years 1991-2023. The right-hand y-axis shows hydrological main values (first occurrence of LLW: 10 October 2022, HHW 23 April 1970). Data: SenMVKU

Summer streamflow in Berlin's river network was lower than average (Fig. 8). Conversely, sewage water discharges were slightly above average (e.g. in 2021 15% higher compared to the long-term summer average) also contributing to a higher sewage water proportion in the river network. The flow direction of the Teltow Canal and the Spree at Spreetunnel downstream of Lake Müggelsee reversed during the summers. In 2019, the Spree experienced reverse flows on 81 days, so that in total 5.8 hm³ of downstream water reached the Müggelsee. In 2020, the Spree had its lowest summer streamflow of 2.15 m³/s in the city centre (around 25% of the long-term average) and 7.03 m³/s (around 40 % of the long-term average) upstream of its confluence with the Havel. Similarly, the Havel had its lowest summer streamflow in 2020 (2.15 m³/s upstream of the mouth of the Spree and 9.04 m³/s downstream, each slightly above one third of the long-term summer average). The driest periods for inflow and outflow occurred in the summers of 2019, 2021 and 2022 so that also the required minimum total outflow of 10 m³/s was not met for consecutive periods of 82, 59 and 52 days, respectively.

Smaller streams (e.g. the Tegeler Fließ and Wuhle rivers) had their lowest summer streamflows in 2022, in some cases 80% below long-term summer averages (fig 8). In 2022 and 2023, many smaller streams ran dry, sometimes for the first time on record (e.g. Wuhle river at Honsfelder Brücke and Erpe river at Krummendammbrücke) or for longer than before (e.g. mostly longer than the dry years 2003 and 2006). At the Fredersdorfer Mühlenfließ River, streamflow only occurred after pronounced rainfall events.

## 4.3 Water resources management and water quality

Berlin's drinking water supply was unrestricted at all times during the 2018-2023 drought. Berlin's water balance, however, and the importance of its individual components was altered during the drought years, so that total inflow and outflow, precipitation and abstractions as well as discharges from thermal power plants were below long-term averages (Fig. 8). Raw water abstractions and sewage water discharge were relatively high, e.g. in summer 2019 5 % and 11 % higher than the long-

term summer averages (Fig. 8). Hence, the proportion of sewage water to surface water in the hydrological network was relatively high and also spatially heterogeneous (Fig. 9a for July-September 2019). For example, the Teltow Canal showed sewage proportions close to 100 % (e.g. at Adlergestell). Return flows of the Spree from the city centre led to high sewage proportions at the Spreetunnel (Fig. 9b). Sewage proportions correlate with detectable trace substance concentrations, as shown in Fig. 9c for the radiographic contrast agent iomeprol. During summer 2019, these concentrations increased locally (Fig. 9d-f).

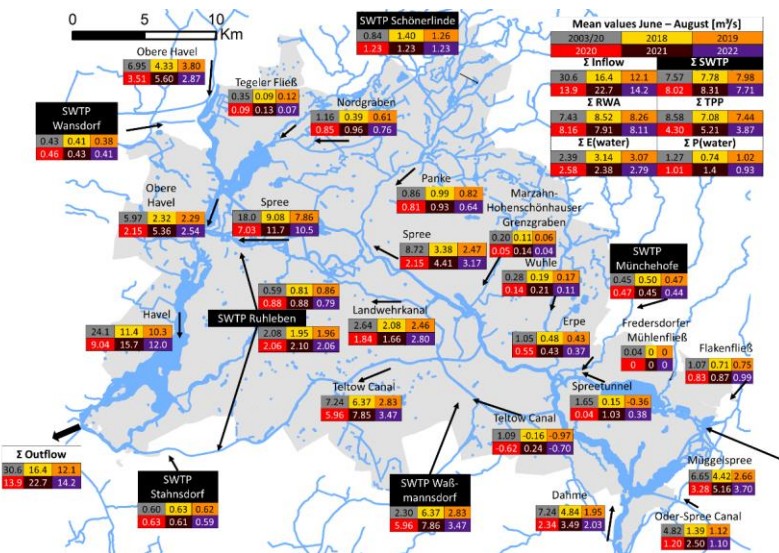

**Figure 8 Streamflow situation in Berlin's hydrological network (BIBER model results) as well as total outflow (sum of streamflow at Tiefwerder and Kleinmachnow shown as small tables in the map for representative cross sections and water balance components of Berlin (top left table): total inflow, total sewage water treatment plant (SWTP) discharges, total raw water abstractions (RWA), total thermal power plant abstractions and discharges (TPP), evaporation (E) and precipitation (P) of the major river network (averaged for JuneAugust 2003-2020 and values for individual years 2018-2023) Arrows indicate (positive) direction of flow. Data: BWB, DWD, SenMVKU, Vattenfall, WSA**

**Table 1: Gauging stations of rivers which temporally ran dry during the hydrological years 2018-2023. Data: SenMVKU**

| Gauging Station | Dry at  W* [cm] | First occurrence | Dry: Number of days per hydrological year | | | | | | | | |
|---|---|---|---|---|---|---|---|---|---|---|---|
| | | | 1991-2020 | 2003 | 2006 | 2018 | 2019 | 2020 | 2021 | 2022 | 2023 |
| Lübars (Tegeler Fließ) | ≤ 0 | 2017 | 0** | 0 | 0 | 0 | 0 | 0 | 0 | 17 | 0 |

| Station | Threshold* | Year | | | | | | | | | |
|---|---|---|---|---|---|---|---|---|---|---|---|
| Röntgental (Panke) | ≤ 28 *** | 2001 | 9 | 93 | 4 | 0 | 0 | 0 | 0 | 17 | 0 |
| Eisenacher Straße (Wuhle) | ≤ 33 | 2008 | 11** | 0 | 0 | 37 | 22 | 0 | 98 | 114 | 134 |
| Wuhletal (Wuhle) | ≤ 29 | 2006 | 18** | 0 | 3 | 61 | 85 | 126 | 97 | 163 | 182 |
| Honsfelder Brücke (Wuhle) | ≤ 9 | 2021 | 0 | 0 | 0 | 0 | 0 | 0 | 22 | 78 | 42 |
| Krummendammbrücke (Erpe) | <= 6 | 2022 | 0 | 0 | 0 | 0 | 0 | 0 | 0 | 82 | 3 |
| Fichtenau (Fredersdorfer Mühlenfließ) | ≤ 17 | 1991 | 82 | 147 | 148 | 160 | 297 | 213 | 237 | 259 | 304 |
| Hegemeisterweg (Fredersdorfer Mühlenfließ) | ≤ 0 | 1998 | 97 | 110 | 133 | 165 | 344 | 333 | 357 | 351 | 364 |

* above gauge datum, ** since observations started, *** before 1994: ≤ 0 cm

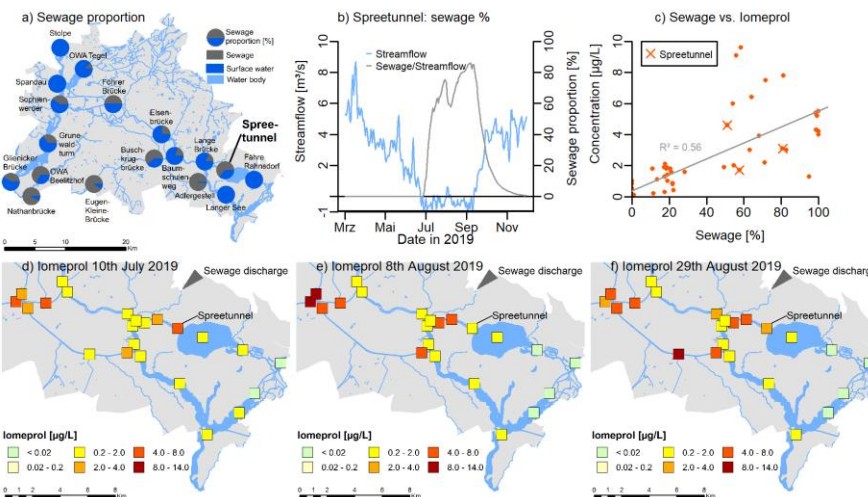

**Figure 9 Wastewater percentage (BIBER model results) and water quality: a) percentage of treated wastewater in streamflow (dark grey) averaged between July and September 2019, in bold: Spreetunnel (cross section where reverse flow to the lake Müggelsee can occur), b) streamflow and percentage of treated wastewater (grey) at the Spreetunnel between March and November 2019, c) relationship between wastewater percentage and concentrations of the radiographic contrast agent iomeprol at 24 monitoring points and 3 sampling dates in 2019 (Spreetunnel highlighted with cross symbols), iomeprol concentrations on 10 July 2019 (d), 8 August 2019 (e), and 29 August 2019 (f). Data: BWB, SenMVKU.**

The water level at the Spandau impoundment, which is crucial for Berlin's water supply as it is home to three waterworks, fell below the operational targets for extended periods (Fig. 10). In the spring of 2018 and 2019, the water level was lowered from winter to summer operational level. In these years, the water level in the Spandau impoundment fell below summer operational level from the end of July until the end of October and reached winter operational level in December. In subsequent years, the water level was not deliberately lowered to summer operational target. Water levels fell below the original summer operational target between July and October 2020, in June 2021, between June and October 2022 and in September and October 2023.

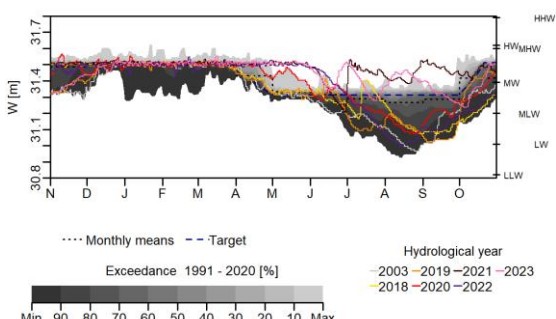

**Figure 10 Water level (W) at the Spandau OP gauging station to Berlin as percentiles for calendar dates and long-term monthly mean values for the hydrological years 1991-2020 and daily values in the hydrological years 2018-2023. The right-hand y-axis shows hydrological main values in the period 2010-2020 (LLW: 10 August 1911, HHW: 22 January 1906), the blue dashed line shows the operation target (winter: 31.514 m, summer: 31.314 m). Data: WSA**

To summarize drought impacts on Berlin, Fig. 11 shows the different variables considered in the hydrological years 2018-2023 in comparison to long-term monthly mean values (1991-2020, for groundwater levels: 2001-2020). Dry periods (illustrated in darker shades) of precipitation and climatic water balance appear almost simultaneously (Fig. 11 a,b) , with soil moisture lagging closely behind (c) and a longer delay for groundwater levels and streamflow (d-j).

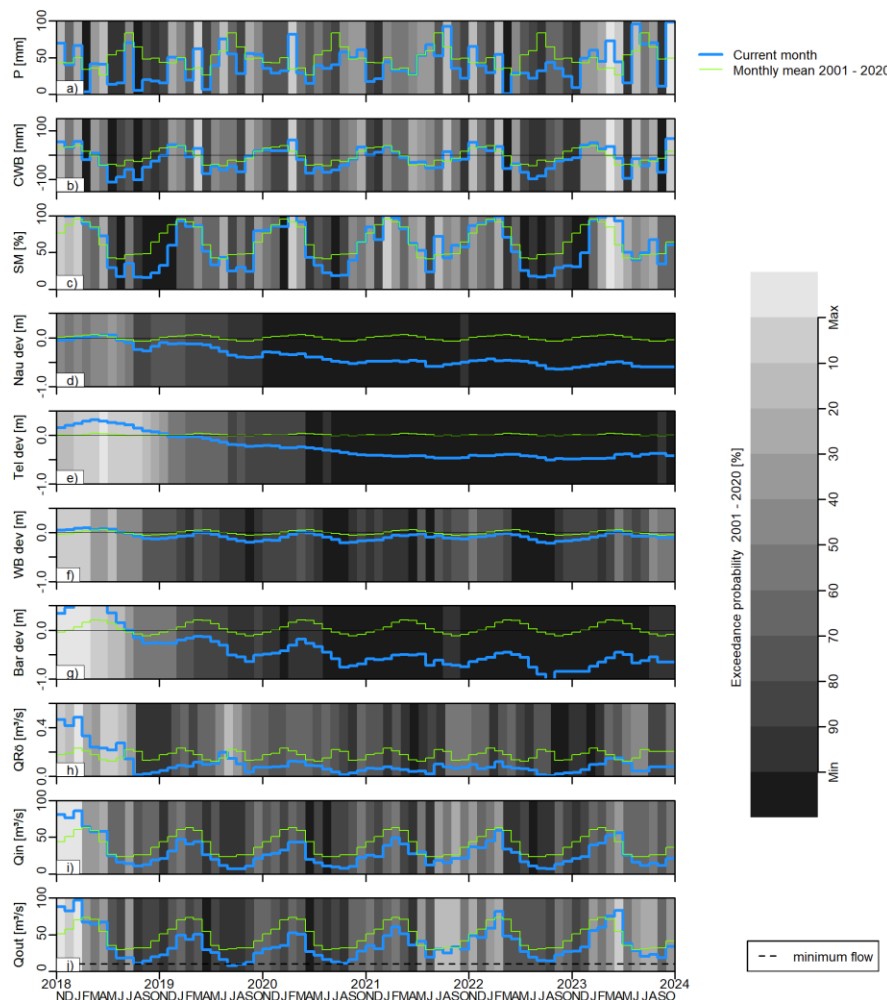

**Figure 11 Summary of drought propagation through Berlin's hydrological cycle for 2018-2023 as monthly values and as percentiles for months compared to long-term monthly mean values of the hydrological years 2001-2020: a) precipitation sum (P), b) sum of the climatic water balance (CWB), c) mean soil moisture (SM), d-g) median deviation**

**from the long-term median groundwater levels of the Nauen Plateau / Grunewald area (Nau dev) (d), Eastern Teltow Plateau (Tel dev) (e), Warsaw-Berlin Glacial valley (WB dev) (f) and Barnim Plateau (Bar dev) (g), h) streamflow of the Panke at Röntgental (QRö), i) inflow to Berlin (Qin) from the tributaries Spree (Große Tränke UP), Dahme (Neue Mühle UP), Oder Spree Canal (Wernsdorf OP) and Obere Havel (Borgsdorf), j) total outflow as sum of Havel (Tiefwerder) and Teltow Canal (Kleinmachnow OP; Qout). Please note that for consistency and due to data availanilty here we compared to 2001-2020 as reference for all variables. Data: DWD, SenMVKU, WSA. 5 Discussion**

**5.1 Propagation and classification of the 2018-2023 drought in Berlin**

The years 2018-2023 years are characterised as meteorological drought, resulting in both a soil moisture drought and hydrological drought, setting new records in hydrometeorological, hydrogeological, and hydrological characteristics. Meteorological drought (precipitation deficit, synchronous low climatic water balance, Fig. 2e, see also supplement S 7a,b) occurred mainly in summer and autumn 2018, autumn 2019 and summer 2020 and 2022. For example, 2022 had the highest mean annual temperature on record, followed by 2020. Both 2018 and 2022 were characterised by the lowest precipitation and the highest and second highest potential evapotranspiration, respectively, and were thus the years with the two lowest values for annual climatic water balance since observations began (Fig. 2e). The region has experienced meteorological droughts in the past, e.g. the year 2003 was exceptionally dry in terms of low precipitation and climatic water balance. However, the sequence of drought years in 2018-2023 exceeded previous observations, as indicated by the strong decrease in the cumulative water balance. The summers generally showed a high influence of evapotranspiration in the region which was also reflected by the isotopic composition (Kuhlemann et al., 2020). During summer days, high evaporation in the Spreewald wetland led to a depletion of water from the Spree and lower outflow from the Spreewald than inflow (AG FGB: Länderübergreifende Auswertung des Niedrigwassers 2018, 2019 und 2020 in den Flussgebieten Spree, Schwarze Elster und Lausitzer Neiße, 2025). Yet, precipitation showed a stronger deviation from long-term averages than evapotranspiration, so that meteorological drought in terms of precipitation is in phase with that expressed by climatic water balance (Fig. 2e, Fig. 11 a,b). Conradt et al. (2023) describe a similar temporal pattern of the 2018-2019 meteorological drought in the German part of the Elbe catchment.

Soil moisture drought slightly lagged meteorological drought (Fig 5, Fig. 11 c) and was most pronounced in summer and autumn 2018, summer 2020 and summer 2022 to winter 2023. As 2017 was a particularly wet year in the Berlin-Brandenburg region (Caldas-Alvarez et al., 2022), 2018 started with relatively high soil moisture, groundwater levels and streamflow. While the onset of soil moisture drought generally lags behind meteorological drought (Zhang et al., 2022), this lag was relatively short in our study due to the dominance of sandy soils. However, soil moisture drought lasted longer than meteorological drought. Soil moisture drought in 2018-2022 further led to agricultural drought in Brandenburg (Brill et al., 2024). Soil moisture deficit also affected Berlin's forests, e.g. in terms of needle loss in pine trees, according to the forest health report of Berlin (SenMVKU: Waldzustandsbericht 2024 des Landes Berlin, 2025). Urban vegetation was also affected, both in terms of reduced vitality of street trees (SenUVK: Straßenbaum-Zustandsbericht Berliner Innenstadt 2020) and extensive damage to trees in historic parks and gardens (Kühn et al., 2024). Both a soil moisture network focusing on urban vegetation in Berlin and a drought monitoring network for the state of Brandenburg (Altdorff et al., 2024) will help to better understand soil moisture drought in the region.

Hydrological drought is visible from 2019 for groundwater and from 2018 for surface water. Since then, groundwater levels on the plateaus have remained in the range of the minima in the 2001-2020 reference period (Fig. 6, Fig. 11 d-g). In line with total water deficit shown by GRACE-FO (Boergens et al., 2020), we also observed more pronounced groundwater deficits in
2019 compared to 2018 despite less severe soil moisture drought which can be attributed to low groundwater recharge in the 2018/2019 winter due to low precipitation. As a comparison to previous drought years, 2003 experienced comparable groundwater decreases as 2018 due to low precipitation and low CWB. However, the 2003 groundwater drought was much shorter as water resources recovered the following year whereas the negative effects of the 2018 drought continued in subsequent years.
The hydrometeorological influence on the hydrological situation is most obvious for the groundwater-fed lake Groß Glienicker See, which showed a decreasing water level synchronous with the decline in cumulative water balance (compare Fig. 7 and Fig. 2e) so that hydroclimatic drivers appear as dominant factor of decreasing lake levels. However, other authors (Somogyvari et al., n.d.) report that also other factors such as large-scale groundwater dynamics as well as vegetation growth and changes in rainwater and sewage water management influenced lake level dynamics of the Groß Glienicker See. Declining lake water
levels in the Berlin-Brandenburg region have already been a concern in recent decades and have in part been attributed to climatic conditions (Kaiser et al., 2015; Natkhin et al., 2012). Streamflow typically shows delayed and buffered response to meteorological drought (Bhardwaj et al., 2020; Van Loon and Laaha, 2015), in Berlin visible for both smaller and larger rivers with streamflow consistently below the long-term averages (Fig. 4, Fig. 8, Fig. 11 7 h-j). Streamflow is impacted by drought effects on different time scales, i.e. lack of rainfall-runoff events and declining groundwater levels, the latter leading to
declining baseflow in the Berlin-Brandenburg region (Warter et al., 2024). In turn, despite a similar hydrometeorological situation, smaller rivers experienced lower flows and dried up for longer periods in 2022 than in 2018 (Tab. 1).
In the recent past, also 2003 and 2006 were characterized by intense hydrological drought in terms of inflows to Berlin, e.g. in terms of low flow indicators as well as drying up of small streams. Yet, similar to meteorological drought, the duration of the hydrological drought in the 2018-2023 period was longer than previously observed. In the past, historical droughts have also
led to low river flows in eastern Germany which have for example been documented by hunger stones in river beds. The drought of 1473 is considered to be one of the most significant heat and drought events of the last millennium due to its duration and spatial extent over large parts of Europe (Hennig, 1904). In the past century, very low flows of the Spree River (so-called water clamps) have been reported e.g. for 1904 (Keller, 1904). These had major impacts on shipping and highlighted the need for reservoir management in the Spree catchment (Keller, 1916). However, continuous hydrological data is not
available for this period, hence a quantitative comparison of historical events with the 2018-2023 drought is not possible.
Water resources management largely influenced the drought propagation to Berlin's major streams but was also affected by the drought in the Spree and Obere Havel catchments and Berlin itself. Due to the intense and prolonged drought in the Spree and Obere Havel catchments, the reservoirs were depleted and not fully filled during the winters (Fig. 3), limiting the ability to mitigate low flows. In turn, the inflows from the Spree and Obere Havel catchments were the lowest on record in terms of
485 lowest mean annual averages and lowest MAM values mostly in either 2019 or 2022 (Fig. 4). Consequently, stream flow in

Berlin was very low and reverse flow occurred in parts of the Spree and the Teltow Canal. Furthermore, sewage proportions were relatively high so that trace substances concentrations were detected. Berlin's water supply was secured despite the unprecedented low streamflow and is a stable system, particularly due to the partially closed circuits.

**5.2 The 2018-2023 drought – a taste of the future?**

Future climate change will further impact the inflows to Berlin and the hydrological processes in Berlin itself. It is expected that future temperatures will rise, whereas future precipitation changes in central Europe are subject to high uncertainties (Evin et al., 2021; Moghim et al., 2022). While 2018 was an extreme drought across Europe, model simulations based on RCP 8.5 suggest that hydrometeorological situations like the summer of 2018 could be common by the middle of the 21$^{st}$ century (Hari et al., 2020; Toreti et al., 2019). Temperature increases might cause more frequent rather than longer soil moisture droughts

(Aalbers et al., 2023). Groundwater levels in Germany might further decline in future (Wunsch et al., 2022), but with high uncertainty as groundwater recharge is extremely difficult to measure and model and therefore to predict. As a result of climatic changes and decreasing mining discharges, future inflows to Berlin are likely to decrease (Pohle et al., 2016). However, uncertainty in precipitation projections results in a wide range of streamflow projections of the Spree (Gädeke et al., 2014; Roers and Wechsung, 2015). It has to be noted, that since the publication of these studies, newer climate scenarios have been

developed and these studies also assumed timetables for the lignite mining phase-out that are now outdated. In 2020, the decision was taken to phase out lignite by 2038 which will lead to an end of mining discharges. Investigating the impacts of more recent climate and mining scenarios and taking appropriate action is currently part of the AG FGB work programme.

The 2018-2023 drought was as severe as climate change impact studies expect for future decades only. For example, the streamflow of the Spree at the Große Tränke UP gauging station was partly lower than expected for the middle of the century

under scenarios of decreasing precipitation (STAR 2K, Orlowsky et al., 2008) and declining mining discharges (compare Fig. 4a and Fig. 8 in Pohle et al. (2016)). In the future, hydrometeorological conditions similar to the 2018-2023 drought would lead to more intense hydrological droughts, because of declining mining discharges and higher evaporation rates from mining pit lakes. Potentially lower groundwater levels (Wunsch et al., 2022) would have stronger consequences for streamflow, with smaller rivers drying up more frequently. Further socioeconomic changes in the region might influence water use (Harrison et

al., 2019), e.g. through increased water demands due to structural change in the Spree catchment and from a growing population in the Berlin/Brandenburg metropolitan region, and hence future streamflow.

For Berlin, it can be assumed that inflow from the Spree and Obere Havel catchments and groundwater recharge will decrease whereas water demand will increase in the future. A number of studies in the past decades and ongoing studies aim to quantify these changes (BAH, 2023; Gädeke et al., 2014; Roers et al., 2016; Stein et al., 2024). Due to the wide range of possible future

changes, it is not advisable to base water resources management planning on a pre-defined selection of scenario outcomes. Therefore, a stress test approach was chosen to assess the vulnerability of Berlin's water resources to declining inflows (SenUVK: Masterplan Wasser Berlin. 1. Bericht, 2025). It was shown that  the water levels in Berlin's impoundments, with the exception of the Spandau impoundment, can be maintained even if future inflows continue to decline . In Berlin's main

river network, water can be retained and recirculated thanks to the water level regulation. In this way, wastewater from sewage treatment plants supports the water levels in the impoundments. However, this leads to an increase in the proportion of wastewater in surface water and the associated increased concentrations of trace substances if no measures (e.g. fourth purification stage) are taken. The quantitative reduction in the inflow and water supply with an increase in water demand, leads to a latent water quality issue, whereby other water quality (combined sewer overflows, etc.) and biological (algae growth, changes in bank structure) aspects must be taken into account or play an important role here (Creutzfeldt et al., 2021).

**5.3 Consequences for water resources management**

The impacts of the 2018-2023 drought on Berlin are the result of a number of factors, the most important of which are the hydrometeorological situation and water resources management in the Spree and Obere Havel catchments and in Berlin itself. However, in a perspective of multi-hazards and their interactions in space and time, the drought situation is also linked to the 2022 Oder environmental disaster, due to which transfers from the Oder to the Oder-Spree Canal were limited resulting in even further reduced inflows to Berlin at Große Tränke UP and Wernsdorf OP. Also, low abstractions and discharges from thermal power plants can not solely be explained by hydrometeorological drivers but are mostly related to the global energy crisis (2021–2023). The impact of water resources management measures during the 2018-2023 drought can be seen at the example of the water levels of the Spandau impoundment. The inflows to the Spandau impoundment (streamflow at Borgsdorf, see Fig. 4d) in 2019 were similar to those in 2020 and 2022. Yet, in 2019 the water levels at Spandau OP in 2019 fell below summer operational target much earlier and remained below this target for a longer period. This can mainly explained by the suspension of the targeted reduction of water levels to summer operational level since 2020 (meaning that around 2 million m³ of water have not been deliberately released in spring) among a number of other measures (Creutzfeldt et al., 2023). To evaluate water resources management measures, the complexity and interlinkages between different influences on Berlin's hydrological system need to be taken into account. It is therefore necessary to analyse the effect of measures not only based on observations, but also using model simulations of combinations of different environmental factors and different management scenarios both in the Spree and Obere Havel catchments and in Berlin using appropriate models.

Water resources management in Berlin and the Spree and Obere Havel catchments will face further challenges as a result of climate change, lignite-mining phase-out and other aspects of regional structural change and population growth in the Berlin/Brandenburg metropolitan region. Sewage discharge already plays a major role in Berlin's hydrological system, noticeable e.g. by the occurrence of trace substances also in drinking water protection areas such as Lake Müggelsee which are linked to reverse flows (Fig. 10, see also Creutzfeldt et al. (2021) and Reith et al. (2024). This issue will become even more relevant in the future as the streamflow of the Spree is likely to decline and reverse flows may occur more often. To prepare for more frequent and more intense droughts in the future, water resources management needs to be adapted both in terms of water quantity as well as water quality. In Berlin, measures for a sustainable future water resources management are being brought together under the framework of the Masterplan Wasser (Masterplan Wasser, SenUVK, 2021). Based on the results of a risk analysis, five areas of action (water management, wastewater infrastructure, water supply, cross-cutting measures and

strategies) were identified, from which 32 specific measures were derived. These include amongst others an optimised sewage water treatment, e.g. removal of trace substances in the sewage water treatment plants. Decentralised rainwater management (e.g. green roofs, infiltration throughs) is expected to locally increase groundwater recharge in Berlin and thus will slightly

counteract the tendency towards potentially declining groundwater recharge under changing climatic conditions. The dominance of inflows for the hydrological system in Berlin rather than of local rainfall-runoff processes, also illustrated by the synchronous pattern of inflow from major tributaries and outflow, underlines the importance of interstate and federal-state collaboration in the regional water resources management. The Water Strategy Capital Region 2050 Brandenburg and Berlin (MLUK: Wasserstrategie Hauptstadtregion 2050, 2025) aims to achieve closer cooperation between these federal statesin the

terms of water supply and wastewater disposal. The interstate collaboration in the Spree and its neighbouring Schwarze Elster and Lusatian Neisse catchments was further strengthened in 2022 when the new mandate of the AG FGB was signed. Staffing the office of the AG FGB in 2024 represents another important milestone in the context of federal cooperation in the Spree catchment. In 2023, the German Federal States of Mecklenburg-Western Pomerania, Brandenburg and Berlin and the Federal Republic of Germany represented by the Waterways and Shipping Agency signed a cooperation agreement on river basin

management in the Obere Havel catchment, which will provide the basis for harmonised management principles in the future. Starting in 2024, the German Federal Institute of Hydrology (Bundesanstalt für Gewässerkunde, BfG) is developing models to provide the technical basis for deriving management principles and objectives for the Obere Havel catchment.

The 2018-2023 drought highlights the need to improve the legal and administrative framework for dealing with droughts as natural hazards. At the European level, Blauhut et al. (2022) proposed a EU drought directive for future drought regulation.

At the national level, a guideline or legal framework for drought management needs to consider, for example, the harmonisation of technical conventions (e.g. definition of usable water supply and indicators of drought/water scarcity). It also needs to guide onthe resolution of conflicts of use and the prioritisation of water uses, taking into account the interests of upstream and downstream users. To that end, it is necessary to look beyond national borders, e.g. to France (Ministre de la transition ecologique et de la cohesion des territoires: Guide circulaire de mise en oeuvre des mesures de restriction des usages de l'eau

en période de sécheresse. À destination des services chargés de leur prescription en métropole et en , 2025) or the Netherlands (Ministerie van Infrastructuur en Waterstaat: Landelijk draaiboek waterverdeling en droogte, 2025).

**6 Conclusion and Outlook**

While each of the years 2018-2023 was exceptionally warm and dry, the consecutive years of drought constitute an unprecedented hydro-meteorological situation for the German capital Berlin since records began. The integrative and

580 interdisciplinary analysis of the 2018-2023 drought illustrates the vulnerability of the state of Berlin to droughts against the background that the region is relatively and characterized by a relatively high water demand complex water resources management. We show that particularly long dry spells cause critical hydrogeological and hydrological situations in Berlin characterised by extremely low soil moisture, groundwater and lake water levels and river flows. While smaller rivers dry up,

larger rivers regulated by weirs such as the Spree partially reverse their flow direction. As a result, the water quality deteriorates in terms of high water temperatures, high levels of sewage percentage and enrichment of point sources. Low water levels also limit the passability of small rivers for fish and aquatic invertebrates. Some heavy rainfalls during the 2018-2023 drought caused combined sewer overflows. Due to low river flows, these were not sufficiently diluted, resulting in oxygen depletion and hitherto fish kills.

Various water management measures to combat the 2018-2023 drought in Berlin and the Spree and Obere Havel catchments included adapted reservoir management, e.g. the suspension of the scheduled reduction of the water level of the Spandau impoundment to summer operational level from 2020 onwards. While the drought situation led to restrictions for some water users such as shipping, Berlin's drinking water supply was not affected.

The 2018-2023 drought may give a taste of future effects of drought intensification and prolongation due to climate change and other factors. Similar hydrometeorological situations in the future will cause even more intense low-flow periods as mining discharges into the Spree continue to decline. Water resources management in Berlin will need to be adapted to meet these challenges, e.g. wastewater management needs to be further improved. The dominance of inflows on Berlin's streamflow situation requires further cooperation between decisions makers at both federal and state level within the Spree and Obere Havel catchments.

Taking into account meteorological, hydrogeological and hydrological variables, but also directly including aspects of water resources management and water quality, our analysis demonstrates drought impacts on a large city in an integrative and multidisciplinary way. Extending the analysis to further monitoring data e.g soil moisture, water used for irrigation of urban vegetation and plant indicators will allow further aspects of drought impacts to be included. This study highlights major aspects of drought and water management and will help to better assess and analyse the impacts of droughts in Berlin-Brandenburg and other regions. Such an analysis can guide water management planning in the region under potentially drier conditions and can possibly be transferred and adapted to study drought impacts on other large cities.

**Code availability**

The analyses are written in R. Code can be made available upon request.

**Data availability**

Hydrogeological and hydrological monitoring data of the state of Berlin are available at https://wasserportal.berlin.de/. Meteorological data has been obtained from the German Weather Service and is available via https://opendata.dwd.de/climate_environment/CDC/. Storage contents of the reservoirs have been obtained from the Landesamt für Umwelt Brandenburg, the Saxon Dam Authority and the Lausitzer und Mitteldeutsche Bergbau-Verwaltungsgesellschaft mbH and are not publicly available. Data on water users, i.e. raw water abstractions, sewage water

treatment plants and thermal power plants have been obtained from Berliner Wasserbetriebe, BWB, and Berliner Energie und Wärme, BEW GmBH, and are not publicly available. These data have been made available on request. Water levels and discharges at the monitoring stations of the Federal Waterways and Shipping Administration (WSA) are in part available via https://www.pegelonline.wsv.de and are partially also shown at https://wasserportal.berlin.de/.

**Author contribution**

IP and BC designed the concept. IP and SZ, JB carried out the data analysis. IP, SZ and BC prepared the manuscript with contributions from JB.

**Competing interests**

The authors declare that they have no conflict of interest.

**Acknowledgements**

We sincerely thank Samar Momin and one anonymous reviewer for their valuable and constructive comments, which helped improve the manuscript. We thank German Weather Service, Landesamt für Umwelt Brandenburg, Saxon Dam Authority, Lausitzer und Mitteldeutsche Bergbau-Verwaltungsgesellschaft mbH, Berliner Wasserbetriebe; Berliner Energie und Wärme GmbH as well as the Federal Waterways and Shipping Administrations for data provision.

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
