# Peer review of "The 2018-2023 drought in Berlin: impacts and analysis of the perspective of water resources management"

_Natural Hazards and Earth System Sciences, 2024_

## Author Comment (AC1)

**RC1: 'Comment on nhess-2024-187', Samar Momin, 17 Oct 2024**

This review is concerned with the paper titled "The 2018-2023 drought in Berlin: impacts and analysis of the perspective of water resources management":

**General Comments:**

The paper presents an in-depth and comprehensive study of the 2018-2023 drought in Berlin, focusing on the hydrometeorological, hydrogeological, and water resource management aspects. It effectively addresses the challenges posed by prolonged droughts and provides an analysis of the effectiveness of water management strategies in Berlin and its surrounding catchments, specifically the Spree and Obere Havel catchments.

The title accurately reflects the content, and the abstract offers a clear summary of the objectives, methodology, and findings. The manuscript is well-structured, with clear sectioning that guides the reader through the complex processes discussed.

The figures and tables provide valuable visual data, supporting the analysis, and are of high quality. The manuscript is also well-referenced, with appropriate credit given to previous studies. Overall, the paper offers significant scientific contributions in the context of water resource management under drought conditions, making it relevant for both regional and broader applications in the face of climate change.

*Authors response:*

*We would like to thank the reviewer for the overall positive assessment of our manuscript. The reviewer's comments provided constructive feedback and valuable suggestions, which we look forward to take into account in the revision process.*

*Please refer to individual comments below (in grey & italics).*

**Specific Comments:**

1. **Comparative Analysis with Other Drought Events**: The study focuses primarily on the 2018-2023 drought. It would help to add (one) more comparison(s) with other significant drought event(s), either in Berlin's history or in other regions with similar climatic conditions.

   *Thank you for this suggestion. In both the results and in the discussion sections, we will go into more detail about previous significant dry years in Berlin and the region (e.g. 1982, 2003, where possible), also to highlight the severity of the 2018-2023 drought in comparison with previous events. We will further include information on pre-mining low-flow periods of the Spree in the introduction and relate to these in the discussion section.*

2. **Future Projections and Climate Change**: The paper does a good job addressing future climate change impacts on water availability. It would help if the authors could elaborate on this section to include more specific projections that could provide clearer guidance for water resource management planning.

*While we strongly agree with the suggestion to include or even propose more specific projections to provide clearer guidance for water resources management planning, we are currently unable to provide such guidance based on existing studies. The main reason for this is the high uncertainty in future precipitation. Developing and evaluating future projections of how climate change might affect water resources in the region is the focus of many ongoing projects. We will add a statement on this and refer to the results of current projects.*

**Technical Comments:**

1. **Grammar and Style**:
   - The manuscript is mostly well-written, but some sentences could be simplified for clarity. Some sentences are overly long and could benefit from being split to improve readability.

   *Thank you for this suggestion. We will pay attention to simplifying our text and shortening long sentences in the revision of our manuscript.*

   - Line 84: 9.7 C **should be 9.7°C**

   *Thanks for pointing this out. We will check our manuscript thoroughly to correct for such errors.*

   - Line 240: "The hydrological summers of 2018 and 2022 showed higher temperatures (17.6 °C and 17.0°°C)" — revise the double "°" symbol.

   *Thanks for pointing this out. We will check our manuscript thoroughly to correct for such errors.*

   -
2. **Figure Captions**:
   - The figures are informative, but the captions could be expanded to better explain the relevance of the data presented. For example, Figures 8 & 9.

   *Thanks for this comment. We will revise our figure captions to include more information. For example in the caption for fig. 8 we will include information on the arrows, tables, colours etc. We well also increase the resolution of fig. 8. For figure 9 we will include information on the arrows and explain the relevance of Spreetunnel.*

   -
3. **In-text Citations**:
   - There are minor inconsistencies in the citation format throughout the manuscript. Ensuring that all references follow a consistent style would improve the presentation of the paper.

   *Thanks for pointing this out. We will check our manuscript thoroughly to correct for such errors.*

- o
4. **Clarification on Data Availability**:
    - o Some of the data sources, are not publicly available. It would be helpful to specify if this data can be accessed under certain conditions or if it remains confidential.

*Thanks for this comment. We will include more information about the accessibility of data sets in a revised version of the manuscript.*

---

## Author Response (AR1)

Dear Editors,

We are grateful for the opportunity to submit a revised version of our manuscript "The 2018-2023 drought in Berlin: impacts and analysis of the perspective of water resources management" for consideration in the special issue "Current and future water-related risks in the Berlin-Brandenburg region" of Natural Hazards and Earth System Sciences.

We confirm that this work is original and has not been published elsewhere, nor is it currently under consideration for publication elsewhere.

We have no conflicts of interest to declare.

We sincerely thank the two reviewers for their conctructive feedback which helped improve the manuscript. Please find our point-by-point response to the reviews below; please note that line numbers refer to the track changes version of the manuscript.

Thank you for your consideration of this manuscript,

Ina Pohle on behalf of all authors

**RC1: 'Comment on nhess-2024-187', Samar Momin, 17 Oct 2024**

This review is concerned with the paper titled "The 2018-2023 drought in Berlin: impacts and analysis of the perspective of water resources management":

**General Comments:**

The paper presents an in-depth and comprehensive study of the 2018-2023 drought in Berlin, focusing on the hydrometeorological, hydrogeological, and water resource management aspects. It effectively addresses the challenges posed by prolonged droughts and provides an analysis of the effectiveness of water management strategies in Berlin and its surrounding catchments, specifically the Spree and Obere Havel catchments.

The title accurately reflects the content, and the abstract offers a clear summary of the objectives, methodology, and findings. The manuscript is well-structured, with clear sectioning that guides the reader through the complex processes discussed.

The figures and tables provide valuable visual data, supporting the analysis, and are of high quality. The manuscript is also well-referenced, with appropriate credit given to previous studies. Overall, the paper offers significant scientific contributions in the context of water resource management under drought conditions, making it relevant for both regional and broader applications in the face of climate change.

*Authors response:*

*We would like to thank the reviewer for the overall positive assessment of our manuscript. The reviewer's comments provided constructive feedback and valuable suggestions, which we look forward to take into account in the revision process.*

*Please refer to individual comments below (in grey & italics).*

**Specific Comments:**

1. **Comparative Analysis with Other Drought Events**: The study focuses primarily on the 2018-2023 drought. It would help to add (one) more comparison(s) with other significant drought event(s), either in Berlin's history or in other regions with similar climatic conditions.

   *Thank you for this suggestion. In both the results and in the discussion sections, we included more details about previous significant dry years in Berlin and the region (e.g. 2003, 2006 where possible), also to highlight the severity of the 2018-2023 drought in comparison with previous events. We included more information on pre-mining low-flow periods of the Spree in the introduction and relate to these in the discussion section.*

   *(Line numbers (track-changes version): 248-271, 321, 342-343, 352, Table 1, 432-495)*

2. **Future Projections and Climate Change**: The paper does a good job addressing future climate change impacts on water availability. It would help if the authors could elaborate on this section to include more specific projections that could provide clearer guidance for water resource management planning.

   *While we strongly agree with the suggestion to include or even propose more specific projections to provide clearer guidance for water resources management planning, we are currently unable to provide such guidance based on existing studies. The main reason for this is the high uncertainty in future precipitation. Developing and evaluating future projections of how climate change might affect water resources in the region is the focus of many ongoing projects. We added a statement on this and refer to the results of current projects.*

   *(Line numbers (track-changes version): 522-527)*

**Technical Comments:**

1. **Grammar and Style**:
   o The manuscript is mostly well-written, but some sentences could be simplified for clarity. Some sentences are overly long and could benefit from being split to improve readability.

   *Thank you for this suggestion. We payed attention to simplifying our text and shortening long sentences in the revision of our manuscript.*
   *(Throughout the manuscript)*

   o Line 84: 9.7 C **should be 9.7°C**

   *Thanks for pointing this out, we corrected this. (Line number (track-changes version): 98)*

o Line 240: "The hydrological summers of 2018 and 2022 showed higher temperatures (17.6 °C and 17.0°°C)" — revise the double "°" symbol.

*Thanks for pointing this out, we corrected this. (Line number (track-changes version): 261)*

o

2. **Figure Captions**:
   o The figures are informative, but the captions could be expanded to better explain the relevance of the data presented. For example, Figures 8 & 9.

*Thanks for this comment. We revised our figure captions to include more information. (Line numbers (track-changes version): 386-393, 397-402, 422-429)*

o

3. **In-text Citations**:
   o There are minor inconsistencies in the citation format throughout the manuscript. Ensuring that all references follow a consistent style would improve the presentation of the paper.

*Thanks for pointing this out. We checked our manuscript thoroughly to correct for such errors.*
*(Throughout)*

o

4. **Clarification on Data Availability**:
   o Some of the data sources, are not publicly available. It would be helpful to specify if this data can be accessed under certain conditions or if it remains confidential.

*Thanks for this comment. We included more information about the accessibility of data sets in a revised version of the manuscript.*
*(Line numbers (track-changes version): 620-624)*

**RC2: 'Comment on nhess-2024-187', Anonymous Referee #2, 01 Nov 2024**

**General comments:**

The study by Pohle et al. comprises a multi-disciplinary and transdisciplinary case study on multiple drought indicators and impacts on different compartments and management territories of a large city, in this study the German capital Berlin. The study analysed the extreme drought conditions during the years 2018-2023 using a wide data spectrum ranging from classical meteorological data, water level of nearby reservoirs and lakes, stream flow data of two large rivers relevant for the city, groundwater series, modelled soil moisture data and sewage water discharge amounts and water quality data.

The multi-data approach alone makes this article to stand out as it rare to find drought

studies that carry out a combined analysis of a whole range of meteorological, hydrological, hydrogeological and surface-water quality data specifically for urban settings. The result chapter presents the deficit aspects of the various compartments in turn, followed by a thorough discussion on the form of drought propagation and a projection for the future, when drought conditions may not be an exceptional and atypical behaviour, but a new normal. The discussion ends with a comprehensive list of how a large city such as the capital Berlin would be affected by such a drastic change of water availability, taking into consideration a wide range of management aspects for drinking water supply, surface water quality deterioration, and the potential need for a drought management plan. I very much liked this study and suggest publication with minor correction.

*We are grateful for the overall positive assessment of our manuscript by the reviewer. The comments are valuable suggestions for improving the manuscript. Please refer to the individual comments below (in grey & italics).*

**Specific comments:**

I have four specific comments, where the authors may consider to implement them to enhance the presentation of the topic:

1. Case study character: you may want to add in the abstract, the introduction and the conclusion that the city of Berlin was selected as a case study to show the diversity of indicators, drought impacts and management questions relevant for large cities. Specifically the abstract and conclusion makes it very difficult to orientate on where this study is 'playing', especially for readers outside of Europe.

   *Thank you for this comment. We introduced our study as an example of drought impacts on a city and added more information in abstract and conclusion about Berlin and its hydrological setting.*
   *(Line numbers (track-changes version): 8-15, 29-31)*

2. Drought propagation: The presentation of the drought propagation across compartments should get a more prominent place in your results or discussion section, and I suggest to move the visualisation S7 from the supplement into the main part of the paper, as the entire section 5.1 is actually refer to it.

   *Thank you for this suggestion. We moved S7 to the main text, it is now included as fig. 11.*
   *(Line numbers (track-changes version): 421-429)*

3. Drought indicators: could you please comment why you haven't used the rather standard rainfall deficit indicator SPI or SPEI to relate the change of meteorological, hydrological etc. droughts? It might be a whole new set of analysis, overlaying the rainfall based deficit analysis and the 'real' data analysis (e.g. SPI-6 for hydrological

drought versus low discharge levels actually measured in the rivers) and beyond the scope of this study.

*We agree it would be interesting to assess droughts also in terms of deficit indicators such as SPI, SPEI, SRI. As also mentioned by the reviewer, this would be a new set of analysis and beyond the scope of this manuscript especially as we wanted to apply a consistent method for the different variables used in this study and also show the actual data, e.g. to allow for comparison of absolute values. Further, some of the data sets are too short for standardization whereas for others standardization might not be meaningful due to various influences.*

4. Urban vegetation and plant water availability: lots of different aspects of drought impacts were mentioned (drinking water supply, ship transport, water quality, cooling for energy production), but water demand for urban green was not included, why?

*Thank you for this comment. We agree that it would be very interesting and important to include urban vegetation, especially in terms of water demand for urban green. Both urban vegetation and forests in Berlin were directly affected by the 2018-2023 drought. Unfortunately, no comparative data on water demand for urban green in Berlin is available yet. A soil moisture network was only launched in 2021 only, so that we decided to include modelled rather than observed soil moisture in the manuscript. A sensor network focusing on the soil water balance of urban trees will be established in the near future. However, drought impacts on Berlin's forests and urban vegetation have been reported. We incorporated these aspects in the discussion part of the manuscript.*

*(Line numbers (track-changes version): 453-457)*

**Technical comments:**

Line 8ff: add that this is a case study of Berlin, and that the period 2018-2023 was a drought period within central Europe and/or Eastern Germany.

*Thank you for this suggestion. We revised the abstract so that it becomes more clear that 2018-2023 was a major drought period in Europe and that we analyze this event in terms of a case study for Berlin.*
*(Line numbers (track-changes version): 8-15)*

Line 39: Berlin's drinking water supply was introduced here quite abruptly; maybe inform first, that drought impacts are studied for this city in general.

*Thank you for pointing this out. We revised the sentence so that it the importance of surface water for Berlin's drinking water supply becomes clearer.*
*(Line numbers (track-changes version): 49-53)*

Line 55: change references to normal standards

*Thank you for pointing this out. We checked the manuscript throughout and corrected the references accordingly.*
*(Throughout)*

**Line 181: specify what you mean with 'for grass and sand'**

*This refers to model results from the German weather service (DWD) under the assumption that land cover is grass and soil is sandy. We revised the sentence for clarification.*
*(Line numbers (track-changes version): 196-197)*

**Line 360: where is the origin of this data described – missing in 3.1? Please also provide detailed references to the other data sets, e.g. WSA does not appear in the list of reference.**

*Thank you for pointing this out. We added more information on these data in section 3.1. As the dataset by WSA is not published as such but has been made available to the authors it is not possible to directly include it in the reference list.*
*(Line numbers (track-changes version): 203-206)*

**Line 360: Figure 8 needs better resolution**

*Thank you for this comment. We changed the resolution of this figure.*
*(Line number (track-changes version): 386)*